# Association Analysis of Single-Cell RNA Sequencing and Proteomics Reveals a Vital Role of Ca^2+^ Signaling in the Determination of Skeletal Muscle Development Potential

**DOI:** 10.3390/cells9041045

**Published:** 2020-04-22

**Authors:** Kai Qiu, Doudou Xu, Liqi Wang, Xin Zhang, Ning Jiao, Lu Gong, Jingdong Yin

**Affiliations:** 1College of Animal Science and Technology, China Agricultural University, No. 2 Yuanmingyuan West Road, Beijing 100193, China; qiukai@cau.edu.cn (K.Q.); BS20183040388@cau.edu.cn (D.X.); wangliqi1224@cau.edu.cn (L.W.); zhangxin0904@cau.edu.cn (X.Z.); jiaoning@cau.edu.cn (N.J.); gonglu@cau.edu.cn (L.G.); 2College of Biological Sciences, China Agricultural University, Beijing 100193, China

**Keywords:** ion homeostasis, Ca^2+^, myogenesis, muscle health, differentiation potential

## Abstract

This study is aimed at exploring the mechanism underlying the homeostasis between myogenesis and adipogenesis in skeletal muscle using a special porcine model with a distinct phenotype on muscle growth rate and intramuscular fat deposition. Differentiation potential of muscle-derived Myo-lineage cells of lean-type pigs was significantly enhanced relative to obese-type pigs, while that of their Adi-lineage cells was similar. Single-cell RNA sequencing revealed that lean-type pigs reserved a higher proportion of Myo-lineage cells in skeletal muscle relative to obese-type pigs. Besides, Myo-lineage cells of the lean-type pig settled closer to the original stage of muscle-derived progenitor cells. Proteomics analysis found that differentially expressed proteins between two sources of Myo-lineage cells are mainly involved in muscle development, cell proliferation and differentiation, ion homeostasis, apoptosis, and the MAPK signaling pathway. The regulation of intracellular ion homeostasis, Ca^2+^ in particular, significantly differed between two sources of Myo-lineage cells. Ca^2+^ concentration in both cytoplasm and endoplasmic reticulum was lower in Myo-lineage cells of lean-type pigs relative to obese-type pigs. In conclusion, a higher proportion and stronger differentiation capacity of Myo-lineage cells are the main causes for the higher capability of myogenic differentiation and lower intramuscular fat deposition. Relative low concentration of cellular Ca^2+^ is advantageous for Myo-lineage cells to keep a potent differentiation potential.

## 1. Introduction

Skeletal muscle is a complex and heterogeneous tissue accounting for approximately 40% of body weight, and its mechanical functions and metabolic roles are critical for animal health [1]. Skeletal muscles originate from the paraxial mesoderm, and myogenesis is initiated with the specification of pre-myogenic progenitors and myoblasts [2]. Muscle development and regeneration absolutely depend on myogenesis, whose impairment is implicated in various muscle dysfunctions, including muscular atrophy, sarcopenia, and muscular dystrophy [3,4]. Besides, excessive ectopic lipid accumulation impairs skeletal muscle protein synthesis [5] and leads to not only the decline of muscle contractile function, but also insulin resistance, obesity, and diabetes [6,7,8]. Therefore, it is of great significance for keeping skeletal muscle function normal to enhance myogenesis and restrain ectopic lipid deposition. However, it remains a challenge in the field of muscle biology yet.

The intramuscular fat (IMF) includes extramyocellular lipid and intramyocellular lipid, and the former stored in adipocytes and dispersed among the muscle fascicles occupies the vast majority [9]. Muscle progenitor cells and their descendant myoblasts are plastic and capable of trans-differentiation into non-myogenic cells, including adipocytes [10]. Nevertheless, fibro-adipogenic progenitors represent the main resident cell population implicated in ectopic adiposity development in human skeletal muscle [11]. Mesenchymal stem cells are widely accepted as the common origin of adipogenic and myogenic precursors in skeletal muscle [12]. Skeletal muscle provides the same environment for the coexistence of both myogenic precursors and adipogenic/fibrogenic precursors [10]. Therefore, exploring the changes associated with the balance between adipogenesis and myogenesis is a key point in understanding the regulative mechanism of skeletal muscle development and aging.

Postnatal growth, homeostasis, and regeneration of skeletal muscle require a dynamic process from muscle stem cells to myofibers, during which cell state and identity change over time [13]. The commitment, differentiation, and formation of skeletal muscle is controlled by a core network of transcription factors, including Pax3, Pax7, and a set of myogenic regulatory factors (MRFs), such as Myf5, MyoD, myogenin, and MRF4 [14]. According to the cellular and molecular complexity of the myogenic compartments, the differentiation stages of muscle stem cells can be subdivided, including muscle progenitor cells, satellite stem cells, satellite cells, myoblasts, and myocytes [2]. The evolutionary emergence and diversification of these cell types can be tracked through the comparative analysis of transcriptome based on their specific sub-functions [15]. Recently, single-cell transcriptomics was successfully applied to afford the granularity required to identify distinct cell types, states, and their dynamics, and provide a rich source of information for describing genetic programs of distinct static cell states [16].

The pig is an excellent model for biomedical research of humans due to similar anatomical, physiological, and genomic characteristics [17]. Genetic selection for thinner subcutaneous backfat thickness in pig industry over decades has led to the emergence of lean-type pigs bred from obese-type pigs, resulting in an elevated capacity of muscle development both in mass and speed, which forms a very specific phenotypic distinction for exploring mechanisms underlying the balance between myogenesis and adipogenesis in muscle, as well as muscle development. Meanwhile, it has been shown in animals that enhanced muscle growth is accompanied by the suppressed IMF deposition [10,18,19]. Therefore, it is meaningful to shed light on the coordination of myogenesis and adipogenesis in muscle for human health using a porcine model. In this study, muscle-derived cells are subjected to single-cell RNA sequencing (scRNA-seq) and proteomics analysis to reveal the relationship between muscle-derived cell spectrum and skeletal muscle growth and development.

## 2. Materials and Methods

### 2.1. Ethics Statement

The study was approved by the Institutional Animal Care and Use Committee of China Agricultural University (ID: SKLAB-B-2010-003) and ensured to be in compliance with the U.K. Animals (Scientific Procedures) Act, 1986 and related guidelines.

### 2.2. Animals and Samples

Three in each of neonatal purebred Laiwu (obese-type) and Yorkshire (lean-type) piglets within 3 days-old were purchased from the Laiwu Pig Original Breeding Farm (Laiwu, China) and Beijing Pig Breeding Center (Beijing, China), respectively. The piglets of the same breed were selected from different litters. After piglets being euthanized, about 1.5 g of muscle from the *longissimus dorsi* over the last rib was sampled, promptly rinsed with 75% ethanol for 3 s, and temporarily stored in PBS (Hyclone, Logan, UT, USA) containing penicillin (100 U/mL) and streptomycin (100 mg/mL) before subsequent experiments.

### 2.3. Preparation of Muscle-Derived Cell Suspension

Single-cell suspension of skeletal muscle was obtained through a series of processes previously described [20]. Briefly, muscle tissue was manually minced and digested for 1 h each with protease (0.17%, Sigma-Aldrich, Louis, MO, USA) and collagenase-type XI (0.15%, Sigma-Aldrich) in a thermostatic shaker (37 °C, 90 r/min). DMEM/F12 supplemented with 10% FBS was used to quench the digestion, and the supernatant of dissociated tissue was filtered successively by 100-μm and 40-μm sterile strainers (BD Biosciences, San Jose, CA, USA). Cells were collected by centrifugation at 400× *g* for 5 min and recovered in growth medium. The cell suspension was laid on ice and immediately used for downstream analyses.

### 2.4. Primary Cell Isolation, Culture, and Differentiation

Based on the preplate technique previously reported by our lab [20], the cell suspension was plated in growth medium in a dish coated with collagen I (Sigma-Aldrich) at 37 °C and 5% CO_2_. In addition, the growth medium is composed of DMEM/F12 (Hyclone), 10% FBS (Gibco-BRL, Carlsbad, CA, USA), 2 mM glutamine (Gibco-BRL), and 5 ng/mL bFGF (Peptech, Burlington, MA, USA). Adherent cells within 2 h were obtained as Adi-lineage cells (Adi), including cells isolated from Laiwu (Adi-L) and Yorkshire (Adi-Y) pigs. Adherent cells between 2 and 72 h were collected as Myo-lineage cells (Myo), including cells from Laiwu (Myo-L) and Yorkshire (Myo-Y) pigs. Myo-lineage cells were further purified by taking the rapidly adhering cells out.

To verify cell differentiation potential, both of Adi-lineage and Myo-lineage cells were exposed to adipogenic and myogenic induction. For adipogenic induction, cells were cultured for 3 days in DMEM/high glucose medium containing 10% FBS, 10 μg/mL insulin, 0.5 mM 1-methyl-3-isobutylmethyl-xanthine, and l μM dexamethasone, and another 5 days in DMEM/high glucose medium containing 10% FBS and 10 μg/mL insulin. The efficiency of adipogenic differentiation was assessed by Oil-red O staining. As for myogenic induction, cells were cultured for 5 days in DMEM/F12 medium containing 2% horse serum. Myotubes were visualized and identified by immunofluorescence staining against myosin, and differentiation index and fusion index were analyzed by ImageJ (v1.45s, National Institutes of Health, Bethesda, MA, USA). Horse serum was purchased from Hyclone Ltd., and other reagents used for induction were from Sigma-Aldrich.

### 2.5. Single-Cell RNA Sequencing

Single-cell suspension was purified by the removal of debris, dead cells, and red blood cells using MACS/Debris Removal Solution (130-109-398, Miltenyi Biotec Inc., Bergisch Gladbach, Germany), Dead Cell Removal Kit (130-090-101, Tissue-Tek, VWR, Radnor, PA, USA), and RBC lysing buffer (R7767, Sigma-Aldrich), respectively. Then, cells were labeled in single-cell barcoded droplets using the 10× genomics 3′ Chromium v2.0 platform (Pleasanton, CA, USA) [21]. The library was prepared as the standard process, and its quality was confirmed by library size (Illumina TapeStation high sensitivity, San Diego, CA, USA), dsDNA quantity (qubit), and amplifiable transcript (KAPA Biosystems, KAPA qPCR analysis, Boston, MA, USA). Resulting libraries were mixed in equimolar fashion and sequenced on an Illumina HiSeq 2500 instrument with “rapid run” mode according to standard 10× genomics protocol.

Sample demultiplexing, barcode processing, and single-cell gene counting were carried out by Cell Ranger Single-Cell Software Suite (v2.1.0, http://10xgenomics.com). Specifically, raw base BCL files were demultiplexed into sample-specific FASTQ files through the Cell Ranger mkfastq pipeline. Then, the FASTQ files were handled individually by the Cell Ranger count pipeline, which aligned cDNA reads to the Sscrofa11.1 reference genome (GCA_000003025.6, Ensembl) via the STAR (2.6.0). Valid cell barcodes (1-Hamming-distance from a list of known barcodes) and unique molecular identifiers (UMIs; not homopolymers and sequencing quality score over 10%) were selected from the aligned reads. Regarding the same gene and the same cell, a UMI with 1-Hamming-distance to another UMI with more reads would be corrected as the latter.

UMI normalization was conducted as previously described [22]. Only genes with at least one UMI count detected in at least one cell were retained, and the library-size of each cell was normalized. Cells with <5% of total UMIs belonging to the mitochondrial genome and >500 detected genes were used for downstream analysis. Data were scaled to shrink the effects of the number of genes detected per cell and the percentage of mitochondrial reads.

Dimension reduction was conducted with the *t*-distributed stochastic neighbor embedding (tSNE) method by Seurat. For the given genes, violin and individual tSNE plots were generated by the Seurat toolkit VlnPlot and FeaturePlot functions, respectively. Pseudotemporal analysis was carried out with the five clusters (Groups 1–5) of Myo-lineage cells by the Monocle R package (http://cole-trapnell-lab.github.io/monocle-release).

### 2.6. Tandem Mass Tag (TMT) Labeling and LC-MS/MS Analysis

Total protein extraction from muscle-derived cell samples was conducted with lysis buffer (0.5% SDS, 8 M urea) and a Halt protease inhibitor cocktail (Thermo Fisher Scientific, Rockford, IL, USA) in the ratio of 100:1. The protein concentration was determined by a BCA Protein Assay Kit (Huaxingbio Science, Beijing, China). According to the manufacturer’s instructions, the protein was digested and then labeled by the 6-plex TMT reagent (Art. No. 90111, Thermo Fisher), specifically as (Myo-L1)-130N, (Myo-L2)-130C, (Myo-L3)-131N, (Myo-Y1)-128C, (Myo-Y2)-129N, and (Myo-Y3)-129C. Finally, all the labeled samples were mixed in equal amounts with loading buffer (5 mM ammonium hydroxide solution containing 2% ACN, pH = 10) and fractionated to increase the proteomic depth by high pH reversed–phase liquid chromatography (ACQUITY Ultra Performance LC, Waters, MA, USA).

A total of 10 fractions of the mixed sample in solvent A (2% ACN and 0.1% FA)were, in turn, injected into a C18 reversed-phase column (75 μm × 25 cm, Thermo Fisher) and then separated with a linear gradient of solvent B (80% ACN and 0.1% FA) by a Thermo Scientific Easy nanoLC 1200. Q-Exactive mass spectrometer (Thermo Fisher) was operated to switch automatically between MS and MS/MS acquisition in the data-dependent mode. The full-scan MS spectra (*m*/*z* 350–1300) with a resolution of 70 K were collected and performed by 20 sequential high energy collisional dissociations (HCD) for MS/MS scans with a resolution of 35 K. Dynamic exclusion of 18 s was used to record the microscan.

### 2.7. Protein Identification and Bioinformatics Analysis

MS/MS data analysis was carried out through Protein Discoverer TM Software 2.1 by mapping to uniprot-proteome-UP000008227-Susscrofa (Pig)-26103s-20170226 fasta. The parameters used for searching protein are shown in Appendix A. Peptide spectral matches were validated based on Q-values at a 1% false discovery rate (FDR). Proteins with fold-change (FC) >1.20 or <0.83 and *p* < 0.05 were considered as differentially expressed proteins (DEPs).

Gene ontology (GO) and Kyoto Encyclopedia of Genes and Genomes (KEGG) enrichment analysis were performed with the DEPs by matching Blast2GO software and KEGG database to the NCBI database, respectively. Significantly enriched GO terms or KEGG pathways were identified by Fisher exact test, and *p* < 0.05 was considered as the threshold.

### 2.8. Quantitative Real-Time PCR Analysis

Total RNA samples were extracted from cell samples by a HiPure Total RNA Mini Kit (Magen, Shanghai, China) and then reverse-transcribed into cDNA by a PrimeScript^TM^ RT reagent Kit with gDNA Eraser (TaKaRa, Kyoto, Japan). RT-qPCR analysis was performed with a quantitative real-time PCR (RT-qPCR) kit (TaKaRa) in an AJ qTOWER 2.2 Real-Time PCR system (Analytik Jena AG, Jena, Germany) according to standard procedures. The primers (Appendix A) designed in this study were verified by the conventional threshold that qRT-PCR cycle number is no more than 3.5, while mRNA is amplified to ten times. GAPDH was taken as an internal control, and all samples were measured in triplicate. The relative gene expression level was calculated by the 2^−ΔΔCt^ method [23].

### 2.9. Western Blot

Protein samples (30 μg) along with a pre-stained protein ladder (Thermo Fisher), were separated on SDS polyacrylamide gels by electrophoresis, and then electro-transferred to a polyvinylidene difluoride (PVDF) membrane (Millipore, Billerica, MA, USA), and blocked for 1 h in TBST (Tris-Buffered saline and Tween-20; 20 mmol/L Tris-Cl, 150 mmol/L NaCl, 0.05% Tween 20, pH 7.4) containing 5% non-fat dry milk. The membranes were then incubated with primary antibodies overnight at 4 °C against GAPDH (CSB-PA00025A0Rb), ABCB1 (CSB-PA11737A0Rb), AKR1B1 (CSB-PA001539LA01HU), ALDH1A1 (CSB-PA001565LA01MO), PRMT3 (CSB-PA018733LA01HU), TRIM5 (CSB-PA874861LA01HU), TSTA3 (CSB-PA00935A0Rb), and YARS2 (CSB-PA903762) (1:500 dilution, CUSABIO, Wuhan, China). Washed three times with TBST, the membranes were incubated for 1 h in the dark with DyLightTM 800-labeled secondary antibodies (1:10,000 dilution, KPL, Gaithersburg, MD, USA). Bands detection was performed by the Odyssey Clx (LI-COR, Lincoln, NE, USA), and the density of bands was quantified using an AlphaImager 2200 (Alpha Innotec, San Diego, CA, USA).

### 2.10. Immunocytochemistry

After fixed with 4% paraformaldehyde for 30 min, cells were permeabilized by 0.2% Trutib X-100 in PBS for 10 min. Blocked for 1 h with 3% BSA/PBS, cells were incubated with primary antibody (anti-myosin, 1:300, M4276, Sigma-Aldrich) overnight and then fluorescein-conjugated secondary antibody (ZF-0311, ZSGB-BIO, Beijing, China) at 1:100 for 1 h. Nuclei were stained with DAPI (62248, Thermo Scientific). Representative images from each slide were acquired by the inverted fluorescence microscope.

### 2.11. Determination of Cellular Ion Concentration

Ca^2+^ concentration in the cytoplasm or endoplasmic reticulum was measured by flow cytometry. Cells were collected, washed, and incubated in 5 μg/mL fluo-3 acetoxymethyl ester (Cayman Chemical, Ann Arbor, MI, USA) or mag-fluo-4-AM (M14206, Thermo Fisher) for 30 min at 37 °C in the dark. Flow cytometry was carried out immediately using FACS Calibur Cyometry and Image Cytometry software (BD FACS Calibur, San Jose, CA, USA). Calcium-bound fluo-3 or mag-fluo-4-AM has an emission maximum of 526 nm, which was quantified by excitation with a 488-nm laser and signals were collected using a 530/30 nm band-pass filter.

*N*-(Ethoxycarbonylmethyl)-6-methoxyquinolinium bromide (MQAE, HY-D0090, MedChemExpress, South Brunswick, NJ, USA) was used to estimate the Cl^−^ influx of cells. The fluorescence of MQAE is collisionally quenched upon interaction with Cl^−^, showing a Cl^−^ concentration-dependent fluorescence decrease. Cells were incubated in Cl^−^-containing buffer supplemented with 5 mmol/L MQAE for 45 min at 37 °C before adding 10.5 mol/L KSCN and 1.75 mmol/L valinomycin to quench the intracellular MQAE fluorescence. The fluorescence of cells was measured immediately by the flow cytometry (excitation, 360 nm; emission, 460 nm).

Debris and dead cells were excluded by forward scatter and side scatter. A total of 20,000 live gated events were acquired for each sample. Mean fluorescence intensity representing relative Ca^2+^ or Cl^−^ concentration was calculated from the live cell population.

### 2.12. Statistical Analysis

Experiment data were analyzed using *t*-test procedures of SAS software (Version 9.3, SAS Institute, Cary, NC, USA). Data are shown as means ± SEMs and considered statistically significant at *p* ≤ 0.05.

## 3. Results

### 3.1. The Differentiation Potential of Muscle-Derived Cells

Adi-lineage cells isolated from obese- and lean-type pigs showed no differences in mRNA expression of adipose-specific genes, such as *C/EBPα* (CCAAT enhancer-binding proteins), *LPL* (adipocyte-specific genes including lipoprotein lipase), and *PPARγ* (peroxisome proliferator-activated receptor gamma) (Figure 1A). Satellite cell marker *Pax7* (paired domain transcription factor 7) and myogenic-specific genes (*Myf5*, myogenic factor 5; *MyoD1*, myoblast determination protein 1; *MyoG*, myogenin; *Mymk*, myomaker) were highly expressed in Myo-Y relative to Myo-L (Figure 1B). As shown in Figure 1C, upon adipogenic induction, skeletal muscle-derived Adi-lineage cells isolated from both obese- and lean-type pigs were differentiated into mature adipocytes with a round shape, accumulating lots of lipids. The amount of lipids stained by Oil-red O showed no differences between the two sources of Adi-lineage cells (Figure 1E). Upon myogenic induction, both Myo-Y and Myo-L were gradually committed to multi-nuclei myotubes (Figure 1D). Notably, the differentiation index and fusion index of Myo-Y at the 2th and 4th day during myogenic differentiation were significantly higher than those of Myo-L (Figure 1F).

### 3.2. Profiling and Trajectory Analysis of Myo-Lineage Cells

ScRNA-seq was performed to identify and profile muscle-derived cells in an unbiased manner derived from newborn obese- and lean-type pigs (Figure 2A, left). A total of 3661 and 4257 single cells from obese- and lean-type pigs were obtained, respectively. Median numbers of genes and UMI detected per cell derived from the obese-type pig were 1166 and 3735, while those from the lean-type pig were 1312 and 3840, respectively. Two datasets obtained from obese- and lean-type pigs, respectively, were analyzed in parallel using the same algorithms. As shown in Figure 3A, unsupervised clustering of the gene expression profiles identified eight cell types in both obese- and lean-type pigs. The predicted cell types, including fibroblasts (expressing *COL1A1*, *GSN*, and *LPL*), PDGFRβ^+^ cells (expressing *CD146* and *PDGFRβ*), hematopoietic cells (expressing *MRC* and *PTPRC*), astrocytes (expressing *GFAP* and *PGP9.5*), endothelial cells (expressing *CDH5*, *FABP4*, and *PECAM1*), and 3 Myo-lineage cell clusters (expressing myogenic-specific genes as shown in Figure 3B), were validated using their known cell-type selective markers as shown in brackets.

As shown in Figure 3B, Myo-lineage cells derived from skeletal muscle of pigs were verified by the highly expressed myogenic-specific genes, such as the marker genes shown in brackets, of mesenchymal cells (*CD29* and *CD44*), satellite cells (*CD82*, *NCAM1*, and *Pax7*), and myocytes (*ACTA1*, *TNNT1*, and *TPM1*), while almost no expression of the marker genes of fibroblasts (*C/EBPα* and *PPARγ*), PDGFRβ^+^ cells, hematopoietic cells (*MRC*), hematopoietic cells (*PTPRC*), astrocytes (*GFAP*), and endothelial cells (*FABP4* and *PECAM1*). To facilitate the understanding of the cellular heterogeneity in Myo-lineage cells (obese-type, 1928 cells, 52.7%; lean-type, 2706 cells, 66.8%) derived from two breeds of pigs (Figure 4A, left), five distinct cell subpopulations were identified as Group 1–5 (Figure 4A, right). The major genes (top-100 selectively expressed in each group) for the classification of Myo-lineage cells were visualized in heat-map (Figure 2B). In particular, Myo-lineage cells derived from obese- and lean type pigs are unevenly distributed in Group 1 (2 vs. 2627), Group 2 (939 vs. 5), Group 3 (165 vs. 23), Group 4 (107 vs. 51), and Group 5 (715 vs. 0). The representative signature genes of each group are shown in Figure 4B, specifically *NPM3* and *INTS6* for Group 1, *RHCG* and *CDKN1C* for Group 2, *CDCA3* and *UBE2C* for Group 3, *SLPI* and *KRT14* for Group 4, and *ACTA1*, *ENO3*, and *MYL1* for Group 5. The detailed information of top-100 selectively expressed genes in each group is listed in Appendix A.

Presumed ancestor–descendent relationships were examined between the five subpopulations of Myo-lineage cells by in silico cell trajectory analyses (Figure 4C). Pseudotemporal analysis predicted that satellite stem cells (Group 2) coming from muscle progenitor cells (Group 1) had two developmental trajectories, including satellite cells (Group 3, cell fate A) and myoblasts (Group 4), and the latter subsequently terminated to be myocytes (Group 5, cell fate B). Part of myoblasts (Group 4) appeared to emerge considerably early accompanied by muscle progenitors (Group 1) and extendedly existed as an intermediary cell type between muscle progenitor cells (Group 1) and satellite stem cells (Group 2). Altogether, it suggests that muscle progenitor cells (Group 1) provide a source of both satellite cells (Group 3) and myocytes (Group 5) passing through satellite stem cells (Group 2).

As shown in Figure 4D, the expression of marker genes representing the different status of myogenic differentiation from progenitors to myocytes obviously differed among the five subpopulations of Myo-lineage cells. To be specific, *CCND2*, encoding cyclin D1 related to cell renewal rate and pluripotency, was highly expressed in muscle progenitor cells (Group 1), satellite stem cells (Group 2), and satellite cells (Group 3) relative to myocytes (Group 5). *Six1/4*, widely confirmed to expressed throughout myogenic differentiation, was detected from all Myo-lineage cells (Groups 1 to 5). *Pax7*, a marker gene of early muscle development, was highly expressed in muscle progenitor cells (Group 1), satellite stem cells (Group 2), and satellite cells (Group 3). *Myf6* and *MyoD1*, muscle-specific transcription factors mainly involved in the early stages of myogenic differentiation, showed higher expression in Groups 2–5 relative to muscle progenitor cells (Group 1). *MyoG*, a transcription factor inducing myogenesis, was highly expressed in myoblasts (Group 4) and myocytes (Group 5). *ACTC1* (actin alpha cardiac muscle 1, a major constituent of muscle contractile apparatus belonging to the actin family) showed high expression in Groups 2–5 relative to Group 1.

### 3.3. Quantitative Mapping of the Proteome of Myo-Lineage Cells

Muscle Myo-lineage cells isolated from obese-and lean-type pigs (*n* = 3) were subjected to proteomics analysis using the TMT labeling (Figure 2A, right). An amount of 6199 proteins were identified and quantitated in Myo-lineage cells. Among them, 181 DEPs were detected, including 134 DEPs upregulated and 47 DEPs downregulated in Myo-Y relative to Myo-L (Appendix A). The relative expression of DEPs between obese- and lean-type pigs was visualized in heat-map (Figure 2C).

In order to verify the reliability of proteomics data, 7 DEPs were randomly selected for Western blot analysis, including ABCB1 (ATP binding cassette subfamily B member 1), AKR1B1 (aldo-keto reductase family 1 member B), ALDH1A1 (aldehyde dehydrogenase 1 family member A1), PRMT3 (protein arginine methyltransferase 3), TRIM5 (tripartite motif containing 5), TSTA3 (tissue-specific transplantation antigen P35B), and YARS2 (tyrosyl-tRNA synthetase 2). As shown in Appendix A, the relative abundances of selected proteins between Myo-L and Myo-Y determined by Western blot were highly consistent with the data of TMT analysis.

The top-10 upregulated and downregulated DEPs are listed in Table 1. Among the top 10 upregulated DEPs, up to four proteins are related to cell proliferation and differentiation, including ALDH1A1, CSRP3 (cysteine and glycine-rich protein 3), EHD3 (EH domain containing 3), and TRIM5. MYH3 (myosin heavy chain 3), and MYLPF (myosin light chain, phosphorylatable, fast skeletal muscle, also known as HUMMLC2B) are components of myosin. CAMK2N2 (calcium/calmodulin dependent protein kinase II inhibitor 2) and TNNT2 (troponin T2, cardiac type) are closely related to cellular Ca^2+^ concentration. ABCB1 mainly functions in various molecules transportation across extra- and intra-cellular membranes. MARC2 (mitochondrial amidoxime reducing component 2) plays an important role in obesity by enhancing lipid synthesis.

Among the top-10 downregulated DEPs, four proteins are involved in the regulation of muscle development and regeneration, including COL15A1 (collagen type XV alpha 1 chain), FBN2 (fibrillin 2), ISLR (immunoglobulin superfamily containing leucine rich repeat), and YARS2. Five DEPs are closely implicated with human diseases, namely as AKR1B1, ARMC9 (armadillo repeat containing 9), PRMT3, TMSB15A (thymosin beta 15a), and TSTA3. Besides, NEMP1 (nuclear envelope integral membrane protein 1), an inner nuclear membrane protein, is involved in neural development.

### 3.4. Bioinformatics Exploration of DEPs Via GO and KEGG Analyses

A total of 181 DEPs were used for GO enrichment analysis, and 24 cellular components terms, 32 molecular functions terms, and 69 biological processes terms were significantly enriched. As shown in Appendix A, DEPs were mainly located in four kinds of cellular components, including extracellular matrix, membrane, cytoplasmic complex, and nucleus. On the basis of their molecular function, DEPs can be mainly classified into three categories, including muscle component, transporter and receptor, and ion homeostasis (Appendix A). As shown in Appendix A, Myo-lineage cells derived from the obese- and lean-type pigs had distinct metabolism patterns of carbohydrate and lipid because 13 biological processes associated with carbohydrate metabolism and 11 ones associated with lipid metabolism were significantly enriched with DEPs. In addition, up to 14 significantly enriched biological processes associated with immune regulation. In this study, we mainly focused on biological processes belonging to muscle structure, apoptosis, cell proliferation and differentiation, ion homeostasis, and MAPK signaling pathway (Appendix A).

As shown in Table 2, five KEGG signal pathways were significantly enriched with 181 DEPs, namely as GABAergic synapse, glycerolipid metabolism, lysosome, phosphatidylinositol signaling system, and valine, leucine, and isoleucine degradation. SLC38A5 (solute carrier family 38 member 5), an amino acid transporter, involved in the pathway of GABAergic synapse was downregulated in Myo-Y. ALDH1B1 (aldehyde dehydrogenase 1 family member B1), participating in glycerolipid metabolism, is closely related to cell self-renewal and differentiation. All three DEPs belonging to the lysosome pathway were upregulated in Myo-Y, including CTSC (cathepsin C), GALNS (galactosamine (*N*-acetyl)-6-sulfatase), and GM2A (GM2 ganglioside activator). INPP1 (inositol polyphosphate-1-phosphatase) catalyzing the removal of phosphate groups in the phosphatidylinositol signaling system was upregulated in Myo-Y. Two enzymes, namely, ALDH1B1 and MCCC2 (methylcrotonoyl-CoA carboxylase 2), were involved in the pathway of branched-chain amino acid degradation. DGKA (diacylglycerol kinase alpha), an upregulated protein participated in both of glycerolipid metabolism and phosphatidylinositol signaling system, plays an important role in the resynthesis of phosphatidylinositols.

According to the above bioinformatics analysis, the functions of DEPs were mainly classified into five categories, including muscle development, ion homeostasis, cell proliferation and differentiation, and apoptosis and gene expression (Appendix A). In addition, typical DEPs interactions among the five categories were shown in Figure 5. RNASEH2B (ribonuclease H2 subunit B) participated in both cell proliferation and apoptosis. Cell proliferation and differentiation is linked to muscle development through FN1 (fibronectin 1) and PALLD (palladin, cytoskeletal associated protein). Ion homeostasis linked to cell proliferation and differentiation through LTBP3 (latent transforming growth factor beta binding protein 3), PLCB1 (phospholipase C beta 1), and TGFBR2 (transforming growth factor beta receptor 2). Ion homeostasis and muscle development were closely connected by MYLPF, GPCPD1 (glycerophosphocholine phosphodiesterase 1), RAP2A (RAP2A, member of RAS oncogene family), PLPP3 (phospholipid phosphatase 3), and MFGE8 (milk fat globule-EGF factor 8). Besides, more remarkable, three key DEPs serving as hubs linked three categories, including muscle development, ion homeostasis, and cell proliferation and differentiation, namely as CSRP3, ENPP1 (ectonucleotide pyrophosphatase/phosphodiesterase 1), and WNT5A (Wnt family member 5A).

### 3.5. Distinct Cellular Ion Regulation between Myo-L and Myo-Y

DEPs connecting ion homeostasis with cell differentiation and muscle development included CSRP3, ENPP1, GPCPD1, LTBP3, MFGE8, MYLPF, PLCB1, PLPP3, RAP2A, TGFBR2, and WNT5A. As shown in Figure 6A, upon myogenic induction (*n* = 6), the mRNA expressions of *CSRP3* and *MYLPF* were significantly increased in Myo-lineage cells, while the mRNA expressions of *GPCPD1*, *PLCB1*, *PLPP3*, *TGFBR2*, and *WNT5A* were significantly decreased. An ScRNA-seq analysis revealed their expressions among the sub-populations of Myo-lineage cells (Figure 6B). For instance, *CSRP3* was highly expressed in myocytes (Group 5), while *GPCPD1*, *LTBP4*, *PLPP1*, and *TGFBR2* showed high expression in muscle progenitor cells, satellite stem cells, satellite cells, and myoblasts (Groups 1–4). *MYLPF* exhibited high expression in satellite stem cells, satellite cells, myoblasts, and myocytes (Groups 2–5). The expressions of *PLCB1* and *WNT5A*, which significantly decreased during myogenic differentiation, showed no visible difference among the five groups.

The expressions of DEPs involved in the regulation of cellular ion concentration were visualized by a heat map (Figure 7A). Most of them, except for the downregulated MT-CO1, SH3RF1, and SLC38A5, were upregulated in Myo-Y as compared with Myo-L, including ANXA5, ATP6V1C1, CAMK2N2, CKB, EHD3, IREB2, S100A16, SLC4A4, SLC4A8, TMEM63B, and TNNT2. The cellular ion concentrations, including cytoplasmic Ca^2+^, endoplasmic reticulum Ca^2+^, and cytoplasmic Cl^−^, were detected through flow cytometry (Figure 7B–D). Ca^2+^ concentrations in resting-state, both in cytoplasm and endoplasmic reticulum, were significantly lower in Myo-lineage cells isolated from lean-type pigs as compared with obese-type pigs (Figure 7E,F). There were no differences in the cytoplasmic Cl^−^ concentration of Myo-lineage cells between the two types of pigs (Figure 7G). The expression of Ca^2+^ transporters located in the endoplasmic reticulum membrane (ATP2A1 and ATP2A2, ATPase sarcoplasmic/endoplasmic reticulum Ca^2+^ transporting 1/2) and cytomembrane (ATP2B1 and ATP2B4, ATPase plasma membrane Ca^2+^ transporting 1/4) remarkably differed among the sub-populations of Myo-lineage cells (Figure 7H). *ATP2A1* was highly expressed in myoblasts (Group 4) and myocytes (Group 5), while the expressions of *ATP2A2*, *ATP2B1* and *ATP2B2* were higher in muscle progenitor cells, satellite stem cells, satellite cells, and myoblasts (Groups 1–4) relative to myocytes (Group 5).

## 4. Discussion

During skeletal muscle development, myogenic and adipogenic-fibrogenic progenitor cells are derived from common origin mesenchymal stem cells (MSCs) [24]. The balance between adipogenic and myogenic differentiation of MSCs ultimately determines muscle traits, including muscle development capacity and IMF content [12,24]. As a well-known lean-type breed, Yorkshire pigs, occupying about 60% of lean meat, are characterized by higher muscle growth rate and lower level of IMF content (<2%) compared with Laiwu pigs, a Chinese indigenous breed of obese-type pigs with about 38% of lean meat and 11.6% of IMF [25,26]. In this study, the Adi-lineage and Myo-lineage cells, both isolated from the skeletal muscle of obese- and lean-type pigs, were certificated with distinct differentiation fate and cannot convert to each other in vitro. The proportions of Adi-lineage cells (Laiwu, 42.97%; Yorkshire, 33.12%) or Myo-lineage cells (Laiwu, 52.7%; Yorkshire, 63.6%) in total muscle-derived cells were different between two types of pigs. The differentiation potential of Adi-lineage cells was similar between obese- and lean-type pigs, while it is notable that differentiation potential of Myo-lineage cells in lean-type pigs was more powerful relative to obese-type pigs. Therefore, we deduced that distinct skeletal muscle traits existing between obese- and lean-type pigs were mainly due to the cell proportion and differentiation potential of Myo-lineage cells rather than those of Adi-lineage cells.

The lineage progression from embryonic progenitors to myotubes or myofibers was realized by going through a series of stages of cellular specification, such as satellite stem cells, satellite cells, myoblasts, and myocytes [2]. ScRNA-seq provides a promising avenue for obtaining a global landscape of organogenesis [16,27], such as revealing remarkable heterogeneity in skeletal muscle cells during developmental processes [28]. In this study, heterogeneity and differentiation dynamics of asynchronous Myo-lineage cells isolated from obese- and lean-type pigs were clearly pictured by single-cell analysis. Myo-lineage cells were classified into five subpopulations that are located at distinct points on developmental trajectories. Distinct characteristics of five subpopulations were clearly verified by their top-100 selectively expressed genes. The overwhelming majority of Myo-lineage cells derived from lean-type pig were categorized into muscle progenitors (Group 1), while the counterpart from obese-type pig was widely scattered over stem satellite cells, satellite cells, myogenic cells, myoblasts and myocytes (Groups 2–5). It indicated that the differentiation stage of Myo-lineage cells in neonatal skeletal muscle significantly differed between two breeds of pigs.

CCND2, a cell cycle protein, plays an important role in enforcing the pluripotent state of stem cells that have the necessity to rapidly expand their population [29]. During lineage progression of pluripotent MSCs from ancestral progenitors to mature muscle fibers, myogenic transcription factors are religiously organized in hierarchical gene expression networks [2]. The sine oculis-related homeobox 1 (Six1) and Six4 were considered as the apex of the genetic regulatory cascade that directed pluripotent progenitors towards to myogenic lineage [30]. Pax7 highly expressed in myogenic satellite cells is essential for myogenesis after birth [31]. MyoD functions downstream from Pax3 and Pax7 in the genetic hierarchy of myogenic regulators [13]. Myf5 and MRF4 (also known as Myf6) act upstream of MyoD to direct embryonic multipotent cells into the myogenic lineage, and myogenin is regarded as myogenic differentiation genes [32]. The expression of actin ACTC1, a major constituent of the contractile apparatus associated with muscle function, is a sign of mature muscle tissues [33]. Therefore, based on the result of trajectory analysis of Myo-lineage cells and dynamic expressions of *CCND2*, *Six1/4*, *Pax7*, *Myf6*, *MyoD1*, *MyoG*, and *ACTC1* among the Myo-lineage cells, we defined these five subpopulations of Myo-lineage cells as muscle progenitor cells, satellite stem cells, satellite cells, myoblasts, and myocytes. Relative to the obese-type pig, Myo-lineage cells isolated from the lean-type pig were closer to the original stage of muscle progenitors in neonates, which implied that differentiation stage of Myo-lineage cells determined the capacity of myogenesis of skeletal muscle, and, the closer to the initial stage of myogenic lineage progression, the more powerful muscle development and the fewer ectopic lipid deposition possessed.

Proteomics analysis proved that cell proliferation and differentiation of Myo-lineage cells significantly differed between obese- and lean-type pigs. In detail, nine enriched biological processes were associated with cell proliferation and differentiation, such as mesodermal to mesenchymal transition involved in gastrulation (GO:0060809), regulation of stem cell/mesenchymal stem cell/myoblast/fibroblast proliferation and differentiation (GO:2000648, GO:0048146, GO:0006269, GO:0002053, GO:0045663, GO:0045599, and GO:2000741), regulation of fat cell differentiation (GO:0045599), as well as cell morphogenesis involved in differentiation (GO:0000904). It was evidenced again that differentiation trajectories of Myo-lineage cells differed between obese- and lean-type pigs. Besides, the global proteomic profiles of Myo-lineage cells revealed that major differences between obese- and lean-type pigs belong to categories that include muscle development, ion homeostasis, apoptosis, and the MAPK signaling pathway. In a previous study, we showed that the MAPK signaling pathway participates in the mediation of myogenic differentiation [34,35]. Thereby, in the present study, we mainly focused on the proteomic landscape among cell proliferation and differentiation, cellular ion homeostasis, and muscle development.

Muscle development was highlighted with 18 DEPs through eight molecular functions and eight biological processes. Among them, five DEPs associated with growth factors were obviously upregulated in Myo-lineage cells of lean-type pigs, including ALS (insulin-like growth factor binding protein acid labile subunit), IGFBP2 (insulin-like growth factor binding protein 2), IGFBP4, LTBP3 (latent transforming growth factor beta binding protein 3), and TGFBR2. Growth factors TGF-β1, IGF-I, IGF-II, bFGF, and IGFBPs, have been well demonstrated to promote myogenic differentiation of muscle progenitors and enhance muscle development [36,37,38]. CLTB (clathrin light chain B), participating in the regulation of clathrin assembly and function [39], showed high expression during myogenesis and muscle regeneration [40]. In this study, CLTB, as well as HUMMLC2B/MYLPF, MYH3, and MYPN (myopalladin), serving as components of myotube, were highly expressed in Myo-lineage cells of lean-type pigs. PALLD can enhance the migration activity of cells by changing the state of the actin cytoskeleton and promote cell differentiation and skeletal muscle maturation [41]. FN1 regulated cell differentiation fate by affecting cytoskeletal status [35]. Both of PALLD and FN1 were upregulated in the Myo-lineage cells of lean-type pigs, while GPCPD1, catalyzing the phosphorylation of choline and thus promoting lipid metabolism [42], was downregulated. Therefore, we deduced that Myo-lineage cells of lean-type pigs are significantly advantaged for muscle development as compared with obese-type pigs, which is consistent with their enhanced myogenic differentiation potential.

Ion homeostasis was vastly demonstrated to be associated with cell metabolism and tissue development [43]. In this study, ion homeostasis was enriched with up to fifty DEPs by ten molecular functions and seven biological processes, such as inorganic anion exchanger activity (GO:0005452), anion: anion antiporter activity (GO:0015301), cellular anion homeostasis (GO:0030002), and regulation of cellular pH (GO:0030641). Among them, 40 DEPs were upregulated and 10 downregulated in Myo-lineage cells of lean-type pigs relative to obese-type pigs. In addition, most of DEPs have catalytic activity, such as YARS2 and ADCY7 (adenylate cyclase 7), whereas their functions on ion homeostasis regulation remain unclear. It is noteworthy that 12 pivotal DEPs linked cellular ion homeostasis with cell differentiation and muscle development. Among them, 7 DEPs, including CSRP3, GPCPD1, MYLPF, PLCB1, PLPP3, TGFBR2, and WNT5A, were verified to be involved in myogenic differentiation. Furthermore, the expressions of CSRP3, GPCPD1, LTBP4, MYLPF, PLPP3, and TGFBR2 were significantly different among the five groups of Myo-lineage cells in this study. Therefore, we deduced that distinct differentiation potential between the two sources of Myo-lineage cells results from their different regulation of cellular ion homeostasis.

Moreover, we annotated DEPs that participated in or functionally depended on the regulation of cellular ion homeostasis, including ANXA5, ATP6V1C1, CAMK2N2, CKB, EHD3, IREB2, MT-CO1, S100A16, SH3RF1, SLC38A5, SLC4A4, SLC4A8, TMEM63B, and TNNT2. For example, ATP6V1C1 is involved in intracellular pH regulation by increasing V-ATPase activity [44]. SLC4A4 and SLC4A8 have been shown to mediate Na^+^-driven Cl^−^/HCO_3_^−^ exchange for regulating intracellular pH [45]. MT-CO1 (mitochondrially encoded cytochrome c oxidase I) binds to Cu^2+^ and plays an important role in cell energy metabolism and ATP production [46]. IREB2 is an ion-concentration-sensitive protein, contributing to maintain intracellular ion balance [47]. Mutational deletion of EHD3 decreased the expression of Na^+^/Ca^2+^ exchange transporter in muscle cells, which in turn leads to an increase of Ca^2+^ concentration in the sarcoplasmic reticulum of myocytes [48]. CKB participates in the maintenance of ion gradients, as associated with Ca^2+^-ATPases of the sarcoplasmic reticulum in the regulation of Ca^2+^ transport and with Na^+^, K^+^-ATPase of the plasma membrane [49]. TMEM63B is an osmosensitive Ca^2+^-permeable channel [50]. S100A16 is a Ca^2+^-binding protein involved in numerous biological processes, such as cell proliferation, differentiation, migration, and apoptosis [51]. TNNT2, a component of the troponin complex, is involved in the regulation of myosin and actin interactions in response to intracellular Ca^2+^ fluctuations [52]. SH3RF1 (SH3 domain containing ring finger 1) is involved in the regulation of Ca^2+^ uptake and metabolism [53]. Therefore, the regulation of intracellular ion homeostasis, Ca^2+^ in particular, is tightly linked to the differentiation of Myo-lineage cells and significantly differed between two sources of Myo-lineage cells.

Ca^2+^, serving as the second messenger, regulates numerous cellular processes at the center of cell signaling [54] and has been shown to participate in the regulation of myogenic differentiation [35,55]. In this study, Ca^2+^ concentration in both cytoplasmic and endoplasmic reticulum was lower in Myo-lineage cells of lean-type pigs relative to obese-type pigs. Accordingly, expression of Ca^2+^ transporters, including *ATP2A1/2* and *ATP2B**1/4*, differed among subpopulations of Myo-lineage cells. Therefore, we deduced that low levels of Ca^2+^ in Myo-lineage cells of lean-type pigs are probably associated with enhanced differentiation potential relative to obese-type pigs.

## 5. Conclusions

We demonstrate with cell trajectory that muscle progenitor cells differentiated to be satellite stem cells, subsequently diverging into satellite cells and myoblasts, and the latter further differentiated to be myocytes in lineage progression. As a result of intense genetic selection for high lean meat yield, remarkable skeletal muscle characteristics of lean-type pigs are mainly established on enhanced myogenesis rather than depressed adipogenesis relative to obese-type pigs. Muscle-derived cells of lean-type newborn pigs were closer to the original developmental stage of muscle progenitor cells as compared with those of obese-type pigs. Furthermore, the divergence of differentiation potential of Myo-lineage cells was closely associated with the distinct regulation of intracellular ion homeostasis, Ca^2+^, in particular. Low concentration of cellular Ca^2+^ was advantageous for Myo-lineage cells to keep a potent differentiation potential. The conclusions fuel contributions to the knowledge of myogenesis, hold value for improving muscle development and regeneration, and provide a new perspective for the treatment of muscle-related diseases.

## Figures and Tables

**Figure 1 cells-09-01045-f001:**
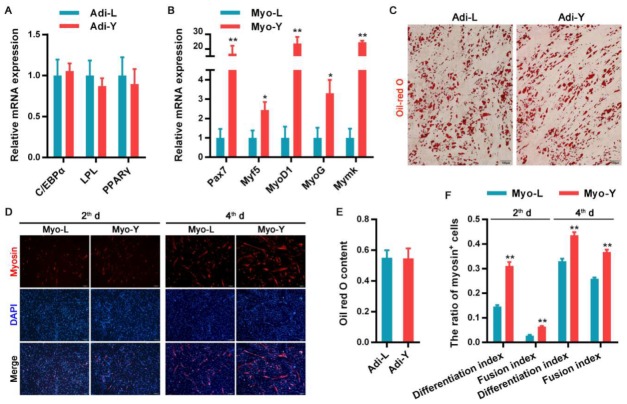
Differentiation potential comparison of Adi-lineage and Myo-lineage cells derived from porcine skeletal muscle between obese- and lean-type pigs (*n* = 3). (**A**) The mRNA expression of adipogenesis-related genes. (**B**) The mRNA expression of myogenesis-related genes. (**C**) Oil-red O staining after adipogenic induction. (**D**) Myosin expression visualized by immunofluorescence at the 2th or 4th day during myogenic induction. (**E**) Oil-red O content quantified by spectrophotometer. (**F**) Differentiation index and fusion index of Myo-lineage cells after differentiation. * means *p* < 0.05, and ** means *p* < 0.01. The magnification of the microscope is 10 × 10. Adi-L and Adi-Y represent Adi-lineage cells derived from skeletal muscle of obese- and lean-type pigs, respectively. Myo-L and Myo-Y represent Myo-lineage cells derived from skeletal muscle of obese- and lean-type pigs, respectively.

**Figure 2 cells-09-01045-f002:**
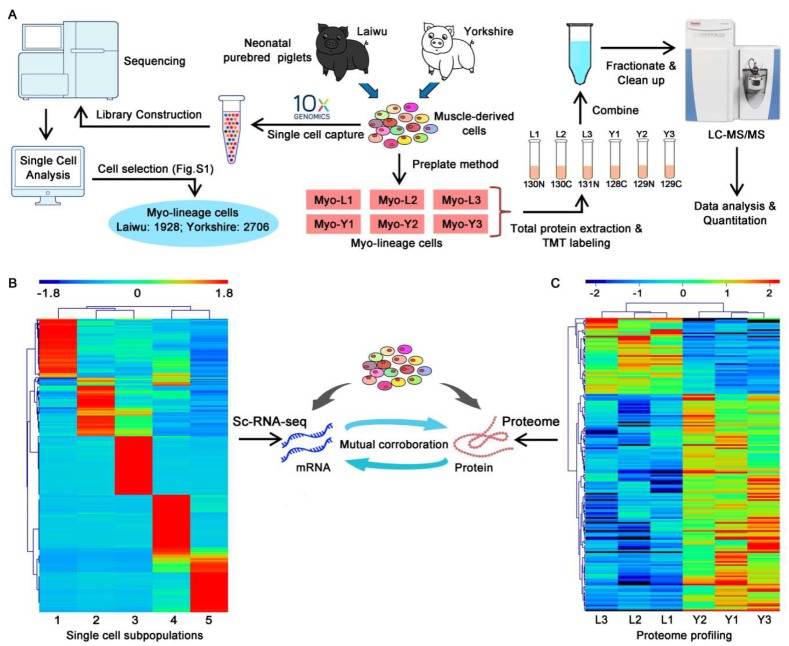
Single-cell RNA sequencing and proteomics analysis of Myo-lineage cells derived from obese- and lean-type pigs. (**A**) Technical process of single-cell RNA sequencing and proteomics analysis. (**B**) The major genes (top-100 selectively expressed in each group) for classification of Myo-lineage cells. (**C**) Hierarchical clustering of differentially expressed proteins. Heat-map was generated by MeV (4.9.0). Myo-L: Myo-lineage cells derived from skeletal muscle of obese-type pigs; Myo-Y: Myo-lineage cells derived from skeletal muscle of lean-type pigs.

**Figure 3 cells-09-01045-f003:**
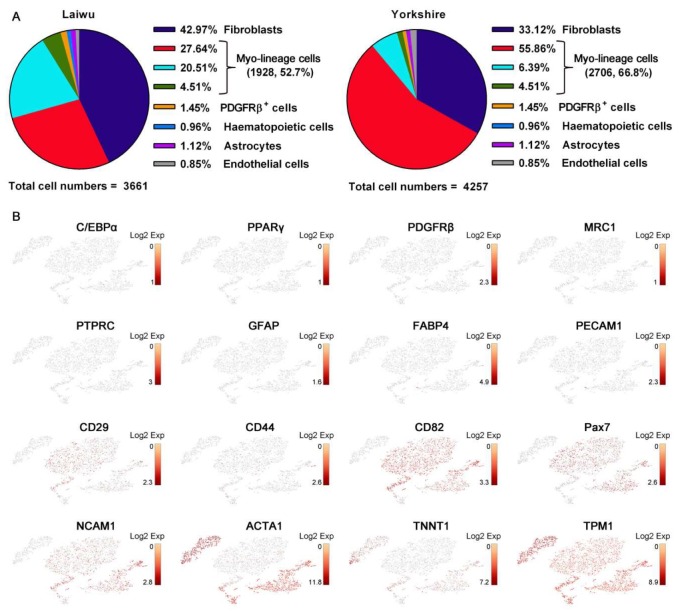
Skeletal muscle cells classified by single-cell RNA sequencing. (**A**) The clusters of muscle-derived cells of obese-or lean-type pigs. (**B**) Individual gene t-distributed stochastic neighbor embedding (tSNE) plots showing the expression levels of cell-type selective markers among Myo-lineage cells.

**Figure 4 cells-09-01045-f004:**
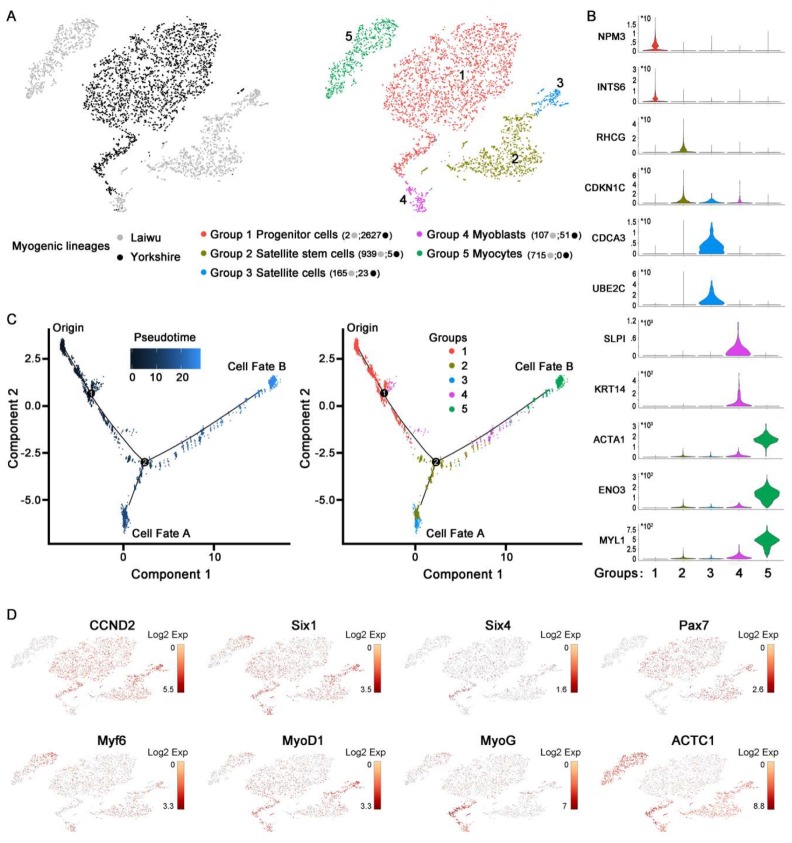
Single-cell RNA expression profiling and cell trajectory analysis delineate the lineage hierarchy of Myo-lineage cells derived from obese- and lean-type pigs. (**A**) Unsupervised clustering of Myo-lineage cells as 3661 from the obese-type pig (Laiwu) and 4257 from the lean-type pig (Yorkshire). (**B**) Violin plots showing the expression levels and distribution of representative marker genes. (**C**) Pseudotemporal cell ordering of Groups 1 to 5 along differentiation trajectories by using Monocle. Pseudotime (arbitrary units) is depicted from dark to light blue (left). Group identities were overlaid on the pseudotime trajectory map (right). (**D**) Individual gene tSNE plots showing the expression levels of genes related to cell differentiation among Myo-lineage cells.

**Figure 5 cells-09-01045-f005:**
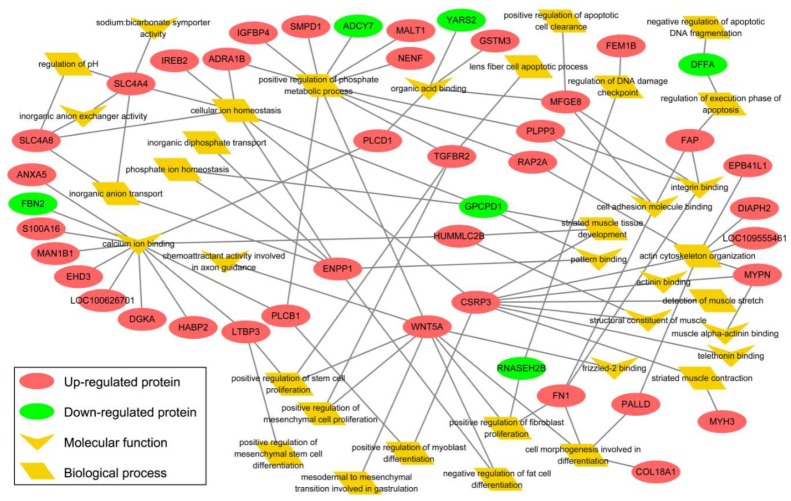
Interaction of differentially expressed proteins of Myo-lineage cells between obese-and lean-type pigs involved in muscle development, ion homeostasis, cell proliferation and differentiation, and apoptosis. The yellow nodes represent one of the molecular functions or biological processes enriched with differentially expressed proteins by GO analysis. The red and green circles mean proteins up-and downregulated in Myo-Y relative to Myo-L, respectively. Networks were drawn and visualized by Cytoscape (v3.2.1). Myo-L and Myo-Y represent Myo-lineage cells derived from skeletal muscle of obese-and lean-type pigs, respectively.

**Figure 6 cells-09-01045-f006:**
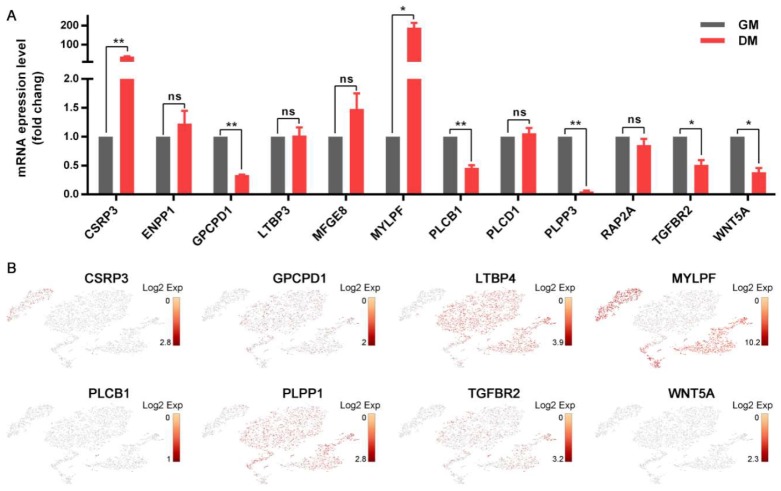
The proteins connecting ion homeostasis with cell differentiation and muscle development. The mRNA expressions of their corresponding genes during myogenic differentiation (**A**, *n* = 6) and among the subpopulations of Myo-lineage cells (**B**). * means *p* < 0.05, and ** means *p* < 0.01. GM, growth medium, representing cells cultured in growth medium before myogenic induction; DM, differentiation medium, representing cells at 4th day during myogenic differentiation cultured in differentiation medium.

**Figure 7 cells-09-01045-f007:**
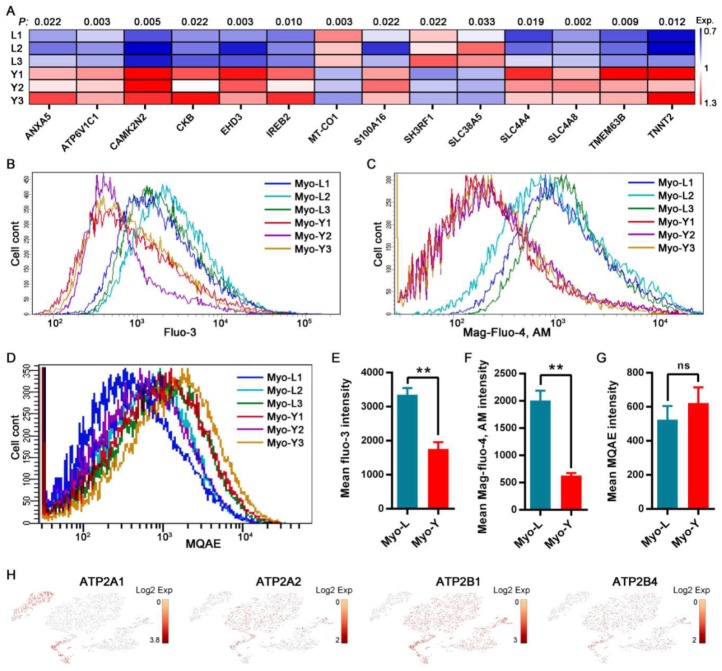
Ion concentration regulation of Myo-lineage cells between obese- and lean-type pigs. (**A**) The differentially expressed proteins involved in the regulation of cellular ion concentration. The scale showed the normalized relative protein expression. (**B**,**E**) The concentration of cytoplasmic Ca^2+^ labeled by Fluo-3 (*n* = 3). (**C**,**F**) The concentration of endoplasmic reticulum Ca^2+^ labeled by mag-fluo-4, AM (*n* = 3). (**D**,**G**) The concentration of cytoplasmic Cl^−^ represented by the fluorescence of *N*-(ethoxycarbonylmethyl)-6-methoxyquinolinium bromide (MQAE) (*n* = 3). (**H**) Individual gene tSNE plots showing the expression levels of Ca^2+^ transports among Myo-lineage cells. * means *p* < 0.05, and ** means *p* < 0.01. Myo-L, Myo-lineage cells derived from skeletal muscle of obese-type pigs; Myo-Y, Myo-lineage cells derived from skeletal muscle of lean-type pigs.

**Table 1 cells-09-01045-t001:** Top-10 upregulated and downregulated proteins.

Protein Name	Gene Name	FC ^1^	*p*-Value
*Upregulated*			
Tripartite motif containing 5	*TRIM5*	4.60	0.001
Aldehyde dehydrogenase 1 family member A1	*ALDH1A1*	2.58	0.036
Calcium/calmodulin dependent protein kinase II inhibitor 2	*CAMK2N2*	2.06	0.005
Myosin heavy chain 3	*MYH3*	2.00	0.011
Myosin light chain, phosphorylatable, fast skeletal muscle	*MYLPF*	1.95	0.012
ATP binding cassette subfamily B member 1	*ABCB1*	1.90	0.035
Mitochondrial amidoxime reducing component 2	*MARC2*	1.75	0.036
Troponin T2, cardiac type	*TNNT2*	1.74	0.012
EH domain containing 3	*EHD3*	1.67	0.004
Cysteine and glycine-rich protein 3	*CSRP3*	1.61	0.043
*Downregulated*			
Immunoglobulin superfamily containing leucine rich repeat	*ISLR*	0.56	0.010
Tyrosyl-trna synthetase 2	*YARS2*	0.58	0.039
Thymosin beta 15a	*TMSB15A*	0.59	0.032
Aldo-keto reductase family 1 member B	*AKR1B1*	0.59	0.025
Fibrillin 2	*FBN2*	0.61	0.007
Tissue specific transplantation antigen P35B	*TSTA3*	0.68	0.000
Armadillo repeat containing 9	*ARMC9*	0.68	0.026
Nuclear envelope integral membrane protein 1	*NEMP1*	0.69	0.011
Protein arginine methyltransferase 3	*PRMT3*	0.69	0.024
Collagen type XV alpha 1 chain	*COL15A1*	0.71	0.014

^1^ FC, fold change, (Myo-Y/Myo-L). Myo-L, Myo-lineage cells derived from skeletal muscle of obese-type pigs; Myo-Y, Myo-lineage cells derived from skeletal muscle of lean-type pigs.

**Table 2 cells-09-01045-t002:** The different expressed proteins involved in the significantly enriched KEGG pathway.

Protein Name	Gene Name	FC ^1^	*p*-Value
*Glycerolipid metabolism*			
Aldehyde dehydrogenase 1 family member B1	*ALDH1B1*	0.80	0.038
Diacylglycerol kinase alpha	*DGKA*	1.22	0.027
*Lysosome*			
Cathepsin C	*CTSC*	1.37	0.040
Galactosamine (*N*-acetyl)-6-sulfatase	*GALNS*	1.37	<0.001
GM2 ganglioside activator	*GM2A*	1.35	0.003
*GABAergic synapse*			
Gamma-aminobutyric acid receptor-associated protein-like 1	*LOC100518837*	1.20	0.043
Solute carrier family 38 member 5	*SLC38A5*	0.83	0.033
*Phosphatidylinositol signaling system*			
Diacylglycerol kinase alpha	*DGKA*	1.22	0.027
Inositol polyphosphate-1-phosphatase	*INPP1*	1.25	0.031
*Valine, leucine and isoleucine degradation*			
Aldehyde dehydrogenase 1 family member B1	*ALDH1B1*	0.80	0.038
Methylcrotonoyl-coa carboxylase 2	*MCCC2*	1.27	<0.001

^1^ FC, fold change, (Myo-Y/Myo-L). Myo-L, Myo-lineage cells derived from skeletal muscle of obese-type pigs; Myo-Y, Myo-lineage cells derived from skeletal muscle of lean-type pigs.

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
