# Peer review of "Association Analysis of Single-Cell RNA Sequencing and Proteomics Reveals a Vital Role of Ca2+ Signaling in the Determination of Skeletal Muscle Development Potential"

_cells, 2020, doi:10.3390/cells9041045_

Round 1
Reviewer 1 Report
The article is in an important topic using cutting edge technologies that are very appropriate to fulfill the experiment. There are some aspects that need to be improved:
- Line 38-40. This sentences should be rewritten to improve the quality of the message it provides.
- Line 41. I would not consider intramuscular fat as ectopic fat. Please justify or remove.
- Line 56. I believe the sentence "Consisting Myf5,..." should be "Consisting in Myf5,...".
- Line 71. Can the authors provide further evidence that "the enhanced muscle growth is accompanied by the suppressed IMF deposition" than just the reference [17]?
- Line 81. Please detail how many animals and technological replicates were used per group.
- Line 100-102. Why did the authors chose the timepoint = 2h to distinguish between Adi- and Myo- lineages?
- Line 226. The terms Myo-L and Myo-Y should be introduced in the materials section.
- Line 249. What are the 2 datasets the authors refer to?
- Line 274-277. This sentence seems incomplete. What was done to "further compare the cellular heterogeneity of Myo-lineage cells" and "we identified 5 distinct myogenic cell subpopulations"?
- Line 577. I think that the statement that the conclusions "provide a potential therapeutic target" is overoptimistic and it should be rewritten.
- Figure 1F: What is Myo-G?
- Figure 2D does not exist.
- Figure 3A: The sum of the percentages of myogenic cell subpopulations in Yorkshire (around 66%) does not match with the one written in the figure (63.6%).
- Figure 6. It is not clear what DM and GM mean in relation to the Methods section.
- Figure 7A. The scale of the expression levels should be detailed.
Author Response
Dear Reviewer,
Thanks for your critical comments and thoughtful suggestions, we wish to reply as follows and have made necessary modifications in the revised manuscript accordingly. All changes made to the text are highlighted using the track changes mode in the revised manuscript. Thanks for your kind consideration.
Comments and Suggestions for Authors
The article is in an important topic using cutting edge technologies that are very appropriate to fulfill the experiment. There are some aspects that need to be improved:
- Line 38-40. This sentences should be rewritten to improve the quality of the message it provides.
Response: We have rewritten it as follows. (Line 40-42)
Therefore, it is of great significance for keeping skeletal muscle function normal to enhance myogenesis and restrain ectopic lipid deposition. However, it remains as a challenge in the field of muscle biology yet.
- Line 41. I would not consider intramuscular fat as ectopic fat. Please justify or remove.
Response: According to the comments, “The ectopic lipid deposition in muscle” was removed. (Line 43)
- Line 56. I believe the sentence "Consisting Myf5,..." should be "Consisting in Myf5,...".
Response: We have revised it into " such as Myf5,...". (Line 59)
- Line 71. Can the authors provide further evidence that "the enhanced muscle growth is accompanied by the suppressed IMF deposition" than just the reference [17]?
Response: According to the reviewer’s suggestion, we supplemented two more references as follows. (Line 74)
Liu, R.; Wang, H.; Jie, L.; Jie, W.; Zheng, M.; Tan, X.; Xing, S.; Cui, H.; Li, Q.; Zhao, G., et al. Uncovering the embryonic development-related proteome and metabolome signatures in breast muscle and intramuscular fat of fast-and slow-growing chickens. BMC Genomics 2017,18, 816.
Yin, H.; Price, F.; Rudnicki, M.A. Satellite cells and the muscle stem cell niche. Physiol. Rev. 2013, 93, 23-67.
- Line 81. Please detail how many animals and technological replicates were used per group.
Response: Three pigs in each group. We have supplemented the information in the manuscript. (Line 86)
- Line 100-102. Why did the authors chose the timepoint = 2h to distinguish between Adi- and Myo- lineages?
Response: The protocol of preplate technique and the time point of 2 h for isolating adipogenic lineage cells employed in the present study was established by our lab in previous study (Sun et al.,2017) according to the Gharaibeh et al. (2008) with some modifications. (Line 105)
Sun, W.; He, T.; Qin, C.; Qiu, K.; Zhang, X.; Luo, Y.; Li, D.; Yin, J.; A potential regulatory network underlying distinct fate commitment of myogenic and adipogenic cells in skeletal muscle. Sci. Rep. 2017, 7, 44133.
Gharaibeh, B.; Lu, A.; Tebbets, J.; Zheng, B.; Feduska, J.; Crisan, M.; Péault, B.; Cummins, J.; Huard, J. Isolation of a slowly adhering cell fraction containing stem cells from murine skeletal muscle by the preplate technique. Nat. Protoc. 2008, 3, 1501–1509.
- Line 226. The terms Myo-L and Myo-Y should be introduced in the materials section.
Response: We have introduced them in the Materials section as follows. (Line 110-112)
Adherent cells within 2 hours were obtained as Adi-lineage cells (Adi), including cells isolated from Laiwu (Adi-L) and Yorkshire (Adi-Y) pigs. Adherent cells between 2 and 72 hours were collected as Myo-lineage cells (Myo), including cells from Laiwu (Myo-L) and Yorkshire (Myo-Y) pigs.
- Line 249. What are the 2 datasets the authors refer to?
Response: The 2 datasets mean the raw data of Single-cell RNA sequencing for obese- and lean-type pigs, respectively. We have defined it in our manuscript as follows. (Line 277-278)
Two datasets obtained from obese- and lean-type pigs, respectively, were analyzed in parallel using the same algorithms.
- Line 274-277. This sentence seems incomplete. What was done to "further compare the cellular heterogeneity of Myo-lineage cells" and "we identified 5 distinct myogenic cell subpopulations"?
Response: We have rewritten the sentence as follows. (Line 302-305)
To facilitate the understanding of the cellular heterogeneity in Myo-lineage cells (obese-type, 1928 cells, 52.7%; lean-type, 2706 cells, 66.8%) between two breeds of pigs (Fig. 4A-left), five distinct subpopulations were identified as group 1-5 (Fig. 4A-right).
- Line 577. I think that the statement that the conclusions "provide a potential therapeutic target" is overoptimistic and it should be rewritten.
Response: We have rewritten it as “provide a new perspective for the treatment of muscle-related diseases”. (Line 616)
- Figure 1F: What is Myo-G?
Response: We have corrected it into “Myo-Y”.
- Figure 2D does not exist.
Response: We updated the Legend of Figure 2 as follows. (Line 289-290)
Figure 2. Single-cell RNA sequencing and proteomics analysis of Myo-lineage cells derived from obese- and lean-type pigs. (A) Technical process of single-cell RNA sequencing and proteomics analysis. (B) The major genes (top-100 selectively expressed in each group) for classification of Myo-lineage cells. (C) Hierarchical clustering of differentially expressed proteins. Heat-map was generated by MeV (4.9.0). Myo-L: Myo-lineage cells derived from skeletal muscle of obese-type pigs; Myo-Y: Myo-lineage cells derived from skeletal muscle of lean-type pigs.
- Figure 3A: The sum of the percentages of myogenic cell subpopulations in Yorkshire (around 66%) does not match with the one written in the figure (63.6%).
Response: Yes, it should be 66.76. We corrected it in the figure and text. (Line 304)
- Figure 6. It is not clear what DM and GM mean in relation to the Methods section.
Response: We have defined it in the Legend of Figure 6 as follows. (Line 450-454)
GM: growth medium, representing cells cultured in growth medium before myogenic induction; DM: differentiation medium, representing cells at 4th day during myogenic differentiation cultured in differentiation medium.
- Figure 7A. The scale of the expression levels should be detailed.
Response: Yes, we detailed it in the legend of Figure 7A as following: The scale showed the normalized relative protein expression. (Line 473-474)
Sincerely,
Jingdong Yin, Ph.D. Professor
College of Animal Science and Technology
China Agricultural University
No. 2 Yuanmingyuan West Road, Beijing 100193, P. R. China
Tel: 8610-62733587; Fax: 8610-62733688
E-mail: yinjd@cau.edu.cn
Reviewer 2 Report
In this manuscript, Dr. Kai Qiu and Collaborators investigated and compared the differentiation capacity and commitment of primary myogenic cell cultures prepared from skeletal muscle samples of two different strains of pigs, namely the obese Laiwu and the lean Yorkshire strains. According to the general target of their project, Authors correctly sampled the neonatal piglets of the two strains.
This comparison was done with a genomic approach, integrating single-cell next-generation sequencing of mRNA (which is per se remarkable and potent as an approach) with proteomics necessarily applied to the bulk cell lineages from which single cells have been isolated for RNA studies.
The results obtained by RNA sequencing and proteomics are interesting and clear and the integration of the two approaches gives a detailed picture of the differences in the molecular signatures between the myogenic cell populations of the lean and obese pigs. I appreciated also the use of single-cell gene expression data to trace the lineage history of myogenic cells.
The Authors finally concentrated their study on specific cell pathways, which comprise genes involved in ion homeostasis of the myogenic cell cultures. Corroborating the proteomic and mRNA expression data with biochemical tests, they were able to discover a significant difference in Calcium ion uptake and metabolism between the two populations of myogenic cells, identifying the principal differentially expressed genes involved in these differences.
Controls of genomic results have been properly conducted (RT-QPCR for RNA and Western blots with antibodies for proteomics), the applied statistical analyses are mainly correct and data are generally sound and solid.
That said, my general impression is that the weaker part of this complex work is the initial characterization of the differentiation capacity of “lean” and “obese” myogenic cell lines, for the reasons that I tried to explain in the following part of my comment. I would suggest that Authors either reconsider and complete the data presented in this part of their work or just decide to omit them: I do not think that, in this case, the general value of the entire manuscript would be diminished or became less comprehensible.
Major remarks
Paragraph 3.1.1.My main concern is the lack of characterization of starting muscle samples. I am afraid that the differences in myogenic differentiation potential between Myo-Y and Myo-L cell lineages could be simply ascribed to the initial number and proportion of myogenic precursors in the muscle biopsies taker from the lean and obese pigs. A clue supporting this potential problem is coming from the classification detected by single-cell RNA sequencing of expanded primary cell lines (Figure 3, panel A) where a 10% difference in myogenic cell and fibroblast subpopulations is measured between Laiwu and Yorkshire, comparable to the differences in the differentiation and fusion indexes shown in Panel F of Figure 1.
To characterize the muscle biopsies, I would suggest the staining of transversal sections with markers specific for satellite cells and fibroblasts and quantification. This should be repeated in both Myo-L and Myo-Y primary culture at time 0 (before induction of differentiation).
In panels of Figure 1, the data of cell culture at time/day 0 (before induction of differentiation) are not presented: they should be shown for useful comparison with measures and staining at different time-points of differentiation.
Figure 2, Panel B. The lower part of the central graph is confusing: the Reader may think that the proteomes were obtained by in-vitro translation of mRNA from muscle-derived cells.
English language
The entire manuscript is punctuated with grammatical and syntax errors, as well as conceptually confused or scientifically wrong sentences that make sometimes very hard the comprehension of the text. I found this problem especially for the Introduction and Discussion sections of the manuscript. The entire manuscript should therefore be checked and in some passages rewritten by native English speaking to whom Authors are requested to explain well what they want to say and present. Just some examples:
Abstract, line 12: “to explore” better “at exploring”
Abstract, line 16: “while their Adi-lineage cells similar” a verb is missing here
Abstract, line 23: “concentration in both of cytoplasm” of should be omitted
Abstract, line 25: “Myo-lineage cells are main causes” are the main causes; “stronger muscle development” better “higher capability of myogenic differentiation”
Introduction, line 32: “in humans and animals” the two terms , in fact what else are humans if not animals?
Introduction, lines 35-36: “muscle diseases including muscular atrophy, sarcopenia” atrophy and sarcopenia are not always due to a disease (e.g. aging, disuse due to sedentary life, etc.)
Introduction, lines 37: “and leads not only the decline” to is missing
Introduction, line 40: “which remains as a challenge in the field of muscle biology yet”
Introduction, line41: “The ectopic lipid deposition in muscle named as intramuscular fat (IMF)” better “The intramuscular fat (IMF) deposits include”
Introduction, line 42-43: “the former stored in adipocytes and dispersed in the myolin (??) bounds occupies the vast majority” this sentence in badly written and hardly comprehensible.
Introduction, line 47: “stem cells were widely accepted” are, not were
Introduction, line 48: “equitable” is an adjective used in ethical context
Introduction, line 49: “to explore” better: exploring
Introduction, line 56: “consisting” better: such as
Introduction, line 58: “including, but not limited to,” I do not understand to which kind of limitation Authors are referring here
Introduction, line 60: “the comparative analysis of genomes” Authors are suggesting here that the genomic DNA is changing during the different stages of muscle stem cells?
Introduction, lines 69-70: “it has been witnessed” it has been shown
Materials and Methods, line 88: “a series of the processes as we previously described” better: a series of procedures previously described”
Materials and Methods, line 97: “the cell suspension was planted” use “plated” instead
Etcetera…………….
Minor remarks
Paragraph 1
Line 32.Please discuss and add reference of the metabolic role of skeletal muscle, an organ which is not only devoted to mechanical functions (contraction and movement).
Paragraph 2.8.
Amplification efficiency parameters for primers listed in the table S2 is lacking. Please, calculate them to make usable those primers to other authors. Moreover, avoid the use of the delta delta Ct method for gene expression level calculations if those will result different from about 100%.
GAPDH is the only transcript used as a control in the Q-RT-PCR experiments. In our experience, this marker is not completely reliable for muscle tissue since it might be differentially expressed according, for example, to fiber composition. How the Authors identified GAPDH as the reference gene in the real time analyses?
Paragraph 2.9.
Please indicate codes for each antibody used in Western Blot analyses.
Paragraph 2.12
The data should not be described as mean +/- SD or SEM, if non parametric statistics are used (Mann-Whitney U test). The data are either parametric or they are not.
Paragraph 3.2.
Figure 3, Panel B. the plots have been obtained from Myo lineage cells of both Laiwu and Yorkshire pigs?
Figure 4, Panel A. In the left part of this Panel it is very hard to distinguish the contribution to clouds density of Laiwu (grey dots) and Yorkshire (black dots) respectively, especially for cells of Group 1. Please, provide numbers or relative percentages of grey and black dots classified in the five Groups. This would quantitatively confirm what Authors claim in the second part of this paragraph (lines 278-280).
I do not understand why Authors cite the figure S1B on line 255.
Author Response
Dear Reviewer,
Thanks for your critical comments and thoughtful suggestions, we wish to reply as follows and have made necessary modifications in the revised manuscript accordingly. The entire manuscript was checked to explain well what we want to present. All changes made to the text are highlighted using the track changes mode in the revised manuscript. Thanks for your kind consideration.
Comments and Suggestions for Authors
In this manuscript, Dr. Kai Qiu and Collaborators investigated and compared the differentiation capacity and commitment of primary myogenic cell cultures prepared from skeletal muscle samples of two different strains of pigs, namely the obese Laiwu and the lean Yorkshire strains. According to the general target of their project, Authors correctly sampled the neonatal piglets of the two strains.
This comparison was done with a genomic approach, integrating single-cell next-generation sequencing of mRNA (which is per se remarkable and potent as an approach) with proteomics necessarily applied to the bulk cell lineages from which single cells have been isolated for RNA studies.
The results obtained by RNA sequencing and proteomics are interesting and clear and the integration of the two approaches gives a detailed picture of the differences in the molecular signatures between the myogenic cell populations of the lean and obese pigs. I appreciated also the use of single-cell gene expression data to trace the lineage history of myogenic cells.
The Authors finally concentrated their study on specific cell pathways, which comprise genes involved in ion homeostasis of the myogenic cell cultures. Corroborating the proteomic and mRNA expression data with biochemical tests, they were able to discover a significant difference in Calcium ion uptake and metabolism between the two populations of myogenic cells, identifying the principal differentially expressed genes involved in these differences.
Controls of genomic results have been properly conducted (RT-QPCR for RNA and Western blots with antibodies for proteomics), the applied statistical analyses are mainly correct and data are generally sound and solid.
That said, my general impression is that the weaker part of this complex work is the initial characterization of the differentiation capacity of “lean” and “obese” myogenic cell lines, for the reasons that I tried to explain in the following part of my comment. I would suggest that Authors either reconsider and complete the data presented in this part of their work or just decide to omit them: I do not think that, in this case, the general value of the entire manuscript would be diminished or became less comprehensible.
Major remarks
Paragraph 3.1.1.My main concern is the lack of characterization of starting muscle samples. I am afraid that the differences in myogenic differentiation potential between Myo-Y and Myo-L cell lineages could be simply ascribed to the initial number and proportion of myogenic precursors in the muscle biopsies taker from the lean and obese pigs. A clue supporting this potential problem is coming from the classification detected by single-cell RNA sequencing of expanded primary cell lines (Figure 3, panel A) where a 10% difference in myogenic cell and fibroblast subpopulations is measured between Laiwu and Yorkshire, comparable to the differences in the differentiation and fusion indexes shown in Panel F of Figure 1.
Response: As the reviewer indicated, the initial number and proportion of myogenic precursors in the skeletal muscle are different between lean and obese pigs, but in the determination and comparison of myogenic differentiation potential, the same amount of Myo-Y and Myo-L cells were used in fact. It has been described in the “Discussion” section (Line 495-497), we explained that distinct skeletal muscle traits existing between obese- and lean-type pigs were mainly due to the cell proportion and differentiation potential of Myo-lineage cells.
To characterize the muscle biopsies, I would suggest the staining of transversal sections with markers specific for satellite cells and fibroblasts and quantification. This should be repeated in both Myo-L and Myo-Y primary culture at time 0 (before induction of differentiation).
Response: As the reviewer suggested, we had done immunofluroence analysis using Cyclin D1, Pax7, Pref 1, and CD82 antibodies, but the quality of photographs was not good enough to provide useful information probably due to low specificity of antibodies against pigs. On the other hand, scRNA-seq is a high reliable technique to reveal the heterogeneity of skeletal muscle cells, and the cell types identified by unsupervised clustering of the gene expression profiles could effectively characterize the cellular composition of muscle tissue.
In panels of Figure 1, the data of cell culture at time/day 0 (before induction of differentiation) are not presented: they should be shown for useful comparison with measures and staining at different time-points of differentiation.
Response: Myo-lineage cells did not expressed myosin during proliferation until exposed to myogenic induction, thus we did not provided the data before induction of differentiation.
Figure 2, Panel B. The lower part of the central graph is confusing: the Reader may think that the proteomes were obtained by in-vitro translation of mRNA from muscle-derived cells.
Response: According to the reviewer’s comment, we removed the “Translation” part of the central graph. (Line 285)
English language
The entire manuscript is punctuated with grammatical and syntax errors, as well as conceptually confused or scientifically wrong sentences that make sometimes very hard the comprehension of the text. I found this problem especially for the Introduction and Discussion sections of the manuscript. The entire manuscript should therefore be checked and in some passages rewritten by native English speaking to whom Authors are requested to explain well what they want to say and present. Just some examples:
Abstract, line 12: “to explore” better “at exploring”
Response: We replaced it as the reviewer suggested. (Line 12)
Abstract, line 16: “while their Adi-lineage cells similar” a verb is missing here
Response: We revised it as the reviewer suggested.
Abstract, line 23: “concentration in both of cytoplasm” of should be omitted.
Response: We omitted it. (Line 23)
Abstract, line 25: “Myo-lineage cells are main causes” are the main causes; “stronger muscle development” better “higher capability of myogenic differentiation”
Response: We revised them as the reviewer suggested. (Line 25-26)
Introduction, line 32: “in humans and animals” the two terms, in fact what else are humans if not animals?
Response: According to the comments, we revised it into “animal health”. (Line 33)
Introduction, lines 35-36: “muscle diseases including muscular atrophy, sarcopenia” atrophy and sarcopenia are not always due to a disease (e.g. aging, disuse due to sedentary life, etc.)
Response: According to the comments, “muscle diseases” was revised as “muscle dysfunctions”. (Line 37)
Introduction, lines 37: “and leads not only the decline” to is missing
Response: We checked it as the reviewer suggested. (Line 39)
Introduction, line 40: “which remains as a challenge in the field of muscle biology yet”
Response: We revised it into “ However, it remains as a challenge in the field of muscle biology yet.”. (Line 42)
Introduction, line41: “The ectopic lipid deposition in muscle named as intramuscular fat (IMF)” better “The intramuscular fat (IMF) deposits include”
Response: According to the comments, “ectopic lipid deposition in muscle, named as” was removed. (Line 43)
Introduction, line 42-43: “the former stored in adipocytes and dispersed in the myolin (??) bounds occupies the vast majority” this sentence in badly written and hardly comprehensible.
Response: We have rewritten it as “the former stored in adipocytes and dispersed among the muscle fascicles occupies the vast majority”. (Line 45)
Introduction, line 47: “stem cells were widely accepted” are, not were
Response: Yes, we corrected it. (Line 49)
Introduction, line 48: “equitable” is an adjective used in ethical context
Response: We replaced it with “same”. (Line 50)
Introduction, line 49: “to explore” better: exploring
Response: We revised it as the reviewer suggested. (Line 52)
Introduction, line 56: “consisting” better: such as
Response: We revised it as the reviewer suggested. (Line 59)
Introduction, line 58: “including, but not limited to,” I do not understand to which kind of limitation Authors are referring here
Response: We have rewritten it as “including muscle progenitor cells, satellite stem cells, satellite cells, myoblasts, myocytes, etc.”. (Line 61-62)
Introduction, line 60: “the comparative analysis of genomes” Authors are suggesting here that the genomic DNA is changing during the different stages of muscle stem cells?
Response: We revised “genomes” into “transcriptome”. (Line 63)
Introduction, lines 69-70: “it has been witnessed” it has been shown
Response: We revised it as the reviewer suggested. (Line 73)
Materials and Methods, line 88: “a series of the processes as we previously described” better: a series of procedures previously described”
Response: We revised it as the reviewer suggested. (Line 95)
Materials and Methods, line 97: “the cell suspension was planted” use “plated” instead
Etcetera…………….
Response: We replaced it as the reviewer suggested. (Line 106)
Minor remarks
Paragraph 1
Line 32.Please discuss and add reference of the metabolic role of skeletal muscle, an organ which is not only devoted to mechanical functions (contraction and movement).
Response: According to the comments, we added the corresponding reference and rewritten it as follows. (Line 32-34)
Skeletal muscle is a complex and heterogeneous tissue accounting for approximately 40% of body weight and its mechanical functions and metabolic roles are critical for animal health [1].
- Goodpaster, B.H.; Sparks, L.M. Metabolic Flexibility in Health and Disease. Cell Metab. 2017, 25, 1027-1036.
Paragraph 2.8.
Amplification efficiency parameters for primers listed in the table S2 is lacking. Please, calculate them to make usable those primers to other authors. Moreover, avoid the use of the delta delta Ct method for gene expression level calculations if those will result different from about 100%.
Response: Yes, it is a true that the reviewer concerning over amplification efficiency in Real-time qPCR analysis. However, it has been generally supposed that the amplification efficiency ideally be 100% in studies to simplify the calculation of the relative mRNA expression level of given gene. It had been well discussed in previous study (Livak, et al. 2001). Particularly, primers designed in this study were verified by the conventional threshold that qRT-PCR cycle number is no more than 3.5 while mRNA is amplified to ten times. To respond the reviewer’s comment, we supplemented associated information and reference in the Paragraph 2.8 as follows. (Line 196-198)
The primers (Table S2) designed in this study were verified by the conventional threshold that qRT-PCR cycle number is no more than 3.5 while mRNA is amplified to ten times.
Livak KJ, Schmittgen TD. Analysis of relative gene expression data using real-time quantitative PCR and the 2(-Delta Delta C(T)) Method. Methods. 2001; 25(4):402-408.
GAPDH is the only transcript used as a control in the Q-RT-PCR experiments. In our experience, this marker is not completely reliable for muscle tissue since it might be differentially expressed according, for example, to fiber composition. How the Authors identified GAPDH as the reference gene in the real time analyses?
Response: In this study, the expression level of GAPDH is consistent between treatments of C2C12 cells. Moreover, GAPDH was also widely used as the reference gene in C2C12 related studies.
Paragraph 2.9.
Please indicate codes for each antibody used in Western Blot analyses.
Response: As the reviewer suggested, we supplemented the codes of antibodies. (Line 207-210)
Paragraph 2.12
The data should not be described as mean +/- SD or SEM, if non parametric statistics are used (Mann-Whitney U test). The data are either parametric or they are not.
Response: According to the comments, we re-analyzed our data using t-test, and revised the Paragraph 2.12 as follows. (Line 241-244)
All experiment data were analyzed using t-test procedures of SAS software (Version 9.3, SAS Institute). Data are presented as means ± SEMs. P ≤ 0.05 was considered as the criterion for statistical significance.
Paragraph 3.2.
Figure 3, Panel B. the plots have been obtained from Myo lineage cells of both Laiwu and Yorkshire pigs?
Response: Yes, from both Laiwu and Yorkshire pigs.
Figure 4, Panel A. In the left part of this Panel it is very hard to distinguish the contribution to clouds density of Laiwu (grey dots) and Yorkshire (black dots) respectively, especially for cells of Group 1. Please, provide numbers or relative percentages of grey and black dots classified in the five Groups. This would quantitatively confirm what Authors claim in the second part of this paragraph (lines 278-280).
Response: As the reviewer suggested, we supplemented numbers of grey and black dots in five Groups in Figure 4, and the corresponding “Result” section was rewritten as follows. (Line 307-309)
Myo-lineage cells derived from obese- and lean type pigs are unevenly distributed in group 1 (2 vs 2627), group 2 (939 vs 5), group 3 (165 vs 23), group 4 (107 vs 51), and group 5 (715 vs 0).
I do not understand why Authors cite the figure S1B on line 255.
Response: It is a mistake. We have corrected it into “Fig. 3B”. (Line 284)
Sincerely,
Jingdong Yin, Ph.D. Professor
College of Animal Science and Technology
China Agricultural University
No. 2 Yuanmingyuan West Road, Beijing 100193, P. R. China
Tel: 8610-62733587; Fax: 8610-62733688
E-mail: yinjd@cau.edu.cn
Reviewer 3 Report
I found this paper really valuable as it contributes to the knowledge of myogenesis, regeneration, and provide a potential therapeutic target for muscle-related diseases. The paper really holds value for scientists interested in genetic engineering of farm pigs working on improving muscle development.
Author Response
Dear Reviewer,
Thanks for your high evaluation.
Sincerely,
Jingdong Yin, Ph.D. Professor
College of Animal Science and Technology
China Agricultural University
No. 2 Yuanmingyuan West Road, Beijing 100193, P. R. China
Tel: 8610-62733587; Fax: 8610-62733688
E-mail: yinjd@cau.edu.cn